# Quantitative Evaluation of the Lymph Node Metastases in the Head and Neck Malignancies Using Diffusion-Weighted Imaging and Apparent Diffusion Coefficient Mapping: A Bicentric Study

**Maria Paola Belfiore** [1,*], **Luigi Gallo** [1], **Alfonso Reginelli** [1], **Pasquale Maria Parrella** [1], **Gaetano Maria Russo** [2], **Valentina Caliendo** [1], **Morena Fasano** [3], **Giovanni Ciani** [1], **Raffaele Zeccolini** [1], **Carlo Liguori** [4], **Valerio Nardone** [1] and **Salvatore Cappabianca** [1]

1 Division of Diagnostic of Imaging, Department of Precision Medicine, Campania University "Luigi Vanvitelli", 80138 Naples, Italy
2 Department of Radiology, Santa Maria della Pietà Hospital, 80035 Nola, Italy
3 Division of Medical Oncology, Department of Precision Medicine, Campania University "Luigi Vanvitelli", 80138 Naples, Italy
4 Department of Radiology, Ospedale del Mare Hospital, 80147 Naples, Italy
* Correspondence: mariapaola.belfiore@gmail.com

**Abstract:** This study aimed to determine if diffusion-weighted imaging (DWI) can differentiate between benign and malignant lymph nodes in patients with head and neck cancer. A total of 55 patients with confirmed head and neck cancer and enlarged neck nodes were enrolled and evaluated by two radiologists using a workstation. Lymph nodes were analyzed using 3D regions of interest (ROIs) placed on T2-weighted images and compared to the corresponding DWI images. This study found that DWI and ADC values can be used to assess metastatic lymph nodes in the neck and that the sensitivity, specificity, and AUC of a narrower ROI for recognizing metastases were greater compared to the ADC value of the whole node. The study also found that the size of the ROI affects ADC values. The results suggest that DWI can accurately predict the status of cervical lymph nodes in patients with head and neck cancer and that it may be useful in diagnosing, determining the stage, developing a treatment plan, and monitoring these patients.

**Keywords:** head and neck cancer; lymph nodes metastases; DWI; ADC

## 1. Introduction

Cancers that can develop in the head and neck region, such as those of the mouth and throat, can be challenging to diagnose and determine the stage due to the complex structures and functions in this area. Accurate diagnosis and classification of the biological severity and the stage of the head and neck cancers requires the use of imaging methods [1].

Annually, over 500,000 people worldwide are diagnosed with head and neck cancer (HNC), making it the sixth most prevalent type of cancer. Of patients with HNC, two-thirds have locally advanced disease, meaning they have either a large tumor or cancer that has spread to nearby lymph nodes. Of those diagnosed at a later stage, only 50% live beyond 5 years [2].

Patients with HNC benefit greatly from the non-invasive detection and characterization of lymph node metastases for therapeutic and prognostic purposes, as well as for monitoring the efficacy of treatment. Due to limitations in existing diagnostic imaging technology, a highly reliable imaging method for assessment of lymph nodes is urgently needed [3–6]. Cross-sectional imaging methods characterize lymph nodes in HNC using size and morphologic parameters, which have low sensitivity and specificity. It is difficult to use MRI for detecting metastatic lymph nodes only on the basis of nodal size [7]. The

technique of characterizing based on size is known to have limitations, such as the possibility of metastases existing in non-enlarged lymph nodes and not all enlarged lymph nodes being cancerous [8–10]. To address these limitations, diffusion-weighted imaging (DWI), a subcategory of MRI that does not require an intravenous contrast agent, has emerged as a successful method for distinguishing between healthy and cancerous tissues [11].

DWI works by quantifying the diffusion of water molecules at the micro level within tissues [11–14]. This method has been widely used for analyzing white matter abnormalities, brain malignancies, and diagnosing brain ischemia, and more recently for distinguishing between benign and cancerous tumors outside of the brain [15]. The apparent diffusion coefficient (ADC) is a valuable metric in DWI that assesses tumor cellularity and indicates that tumor regions with lower ADC values are more densely packed with cancer cells. Hence, DWI is considered a reliable imaging biomarker for cancer [16,17]. When comparing metastatic and benign lymph nodes, DWI consistently shows that metastatic lymph nodes have a lower ADC, which reflects the ease of water movement through tissues [18]. The entire lymph node or just the non-necrotic portion can be used as the region of interest for calculating ADC values [19]. ADC values in normal lymph nodes are influenced by the choice of the region of interest, with an increase seen when the hilum is included. The entire lymph node or just the non-necrotic portion can be used as a region of interest (ROI) for calculating ADC values [2,19,20].

The distribution of lymphatic cells and fluid in the cortex and hilum of a lymph node is not consistent. A higher ADC value in the center of the node can be observed because the middle is more flexible than the cortex [21–24]. The normal ADC values in lymph nodes are significantly influenced by the chosen region of interest, with a rise seen when the hilum is included [25].

In this study, we utilized a three-dimensional region of interest analysis to examine if diffusion-weighted imaging (DWI) can differentiate benign from malignant nodes in patients with head and neck cancer.

## 2. Materials and Methods

### 2.1. Patients

This study was approved by the local ethical committee and consent was obtained from all participants. A total of 55 patients with histologically confirmed head and neck cancer and enlarged neck nodes were enrolled between June 2020 and February 2022, with 31 patients being enrolled at the University of Campania Luigi Vanvitelli MRI center and the remaining 24 at the Hospital of the Sea. Participants were required to undergo multiparametric MR imaging, with the biopsy results confirming histological malignancy, and no prior history of neck surgery, biopsy, chemotherapy, or radiotherapy prior to the MRI. Twelve patients were excluded from the analysis for not meeting these criteria. The patients' ages ranged from 55 to 81 years. There were 21 male and 22 female patients, of which 16 patients had left-side involvement, 18 had right-side involvement, and 9 had involvement on both sides. All patients received a definitive pathology diagnosis after undergoing neck lymph node surgery following an MRI. The cancers diagnosed, as determined by histopathology, were 13 cases of pharynx cancer, 6 of hard or soft palate cancer, 5 of tongue cancer, 1 of lip cancer, 8 of parotid cancer, 2 of neck skin cancer, and 8 of larynx cancer.

### 2.2. MRI Imaging and Protocol

The MRI scans were conducted in two different facilities: the University of "Campania Luigi Vanvitelli" which used a General Electric Signa Voyager 1.5T machine and at the "Hospital of the Sea" which used a Siemens MAGNETOM Amira 1.5T system. The MRI sequences included T1 sagittal (voxel size $1.0 \times 1.0 \times 4.0$ mm, distance factor 10%, FoV 250 mm, slice thickness 4 mm, TR 700 ms, and TE 12 ms), STIR coronal (voxel size $1.2 \times 0.9 \times 4.0$ mm, distance factor 30%, FoV 280 mm, slice thickness 4.0 mm, TR 6990 ms, and TE 64 ms) axial view (voxel size $1.2 \times 0.9 \times 4.0$ mm, distance factor 10%, FoV 280 mm, slice thickness 4.0 mm, TR 7530 ms, and TE 64 ms), T2 axial (voxel size $0.9 \times 0.7 \times 3.0$ mm,

distance factor 10%, FoV 230 mm, slice thickness 3.0 mm, TR 7850 ms, and TE 82 ms), T1 fat-saturated axial (voxel size $1.0 \times 0.7 \times 3.0$ mm, distance factor 10%, FoV 230 mm, slice thickness 3.0 mm, TR 1260 ms, and TE 25 ms), coronal view, contrast-enhanced T1 fat-saturated axial, and a diffusion-weighted MRI using spin echoplanar imaging with a TR of 7600 ms, TE of 90 ms, and three b-values (0, 500, and 1000 s/mm$^2$). The field of view was 380 mm, with a matrix of $128 \times 128$, slice thickness of 4.0 mm, and voxel size of $2.0 \times 2.0 \times 4.0$ mm. Careful manual shimming was also performed to improve image quality and reduce fat-shift and distortion artifacts. ADC maps were generated from the diffusion-weighted images (Figure 1).

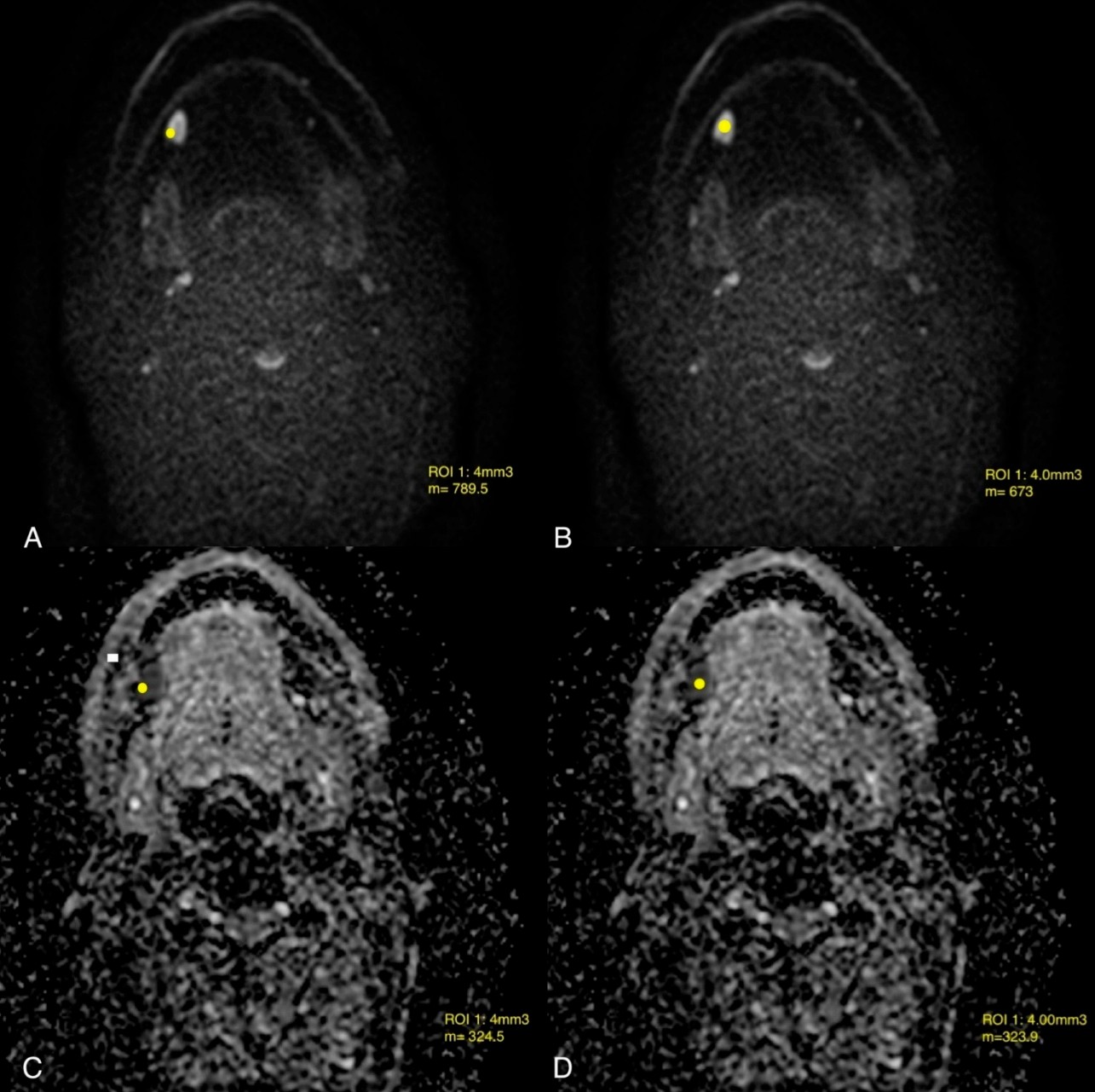

**Figure 1.** Evaluation of a malignant node (station IB) in a 78-year-old male patient with oropharyngeal carcinoma is shown in MR images including T2-weighted sequence (**A**), DWI sequence (**B**), and ADC map with grayscale (**C**) and colorimetric scale (**D**). In the case of a 82-year-old male patient with tongue squamous carcinoma, DWI and ADC maps display the evaluation of a 3D region of interest (ROI) with a size of 0.4 cm$^3$ in both the subcapsular (**A**,**C**) and central (**B**,**D**) areas of the malignant node.

### 2.3. Image Analysis

Two radiologists with extensive experience in head and neck medicine (one with 30 years and one with 10 years of experience) evaluated the MR images using a workstation (AW Server GE Healthcare, Italy). Lymph nodes with necrosis or a diameter between 15 and 9 mm were excluded from the analysis. If a patient had multiple metastatic lymph nodes, the largest one was selected for analysis. The volume of the lymph node was calculated by manually outlining it and inputting the coordinates into the workstation software. The 3D regions of interest (ROIs) were placed manually in the selected lymph nodes using T2-weighted images to determine the size and location of the ROI. The ROIs were then compared to the corresponding DWI images to create 3D ROIs. Only one slice was used to measure the ADC values, and the slice with the largest node diameter was chosen to minimize partial volume artifacts. The ROI in the subcapsular area of both benign and malignant nodes was optimized to 0.4 cm$^3$ and the mean ADC was determined and compared between the two types of lymph nodes. The unit of measure for ADC values was N $\times 10^3$ mm$^2$/s (Figure 2).

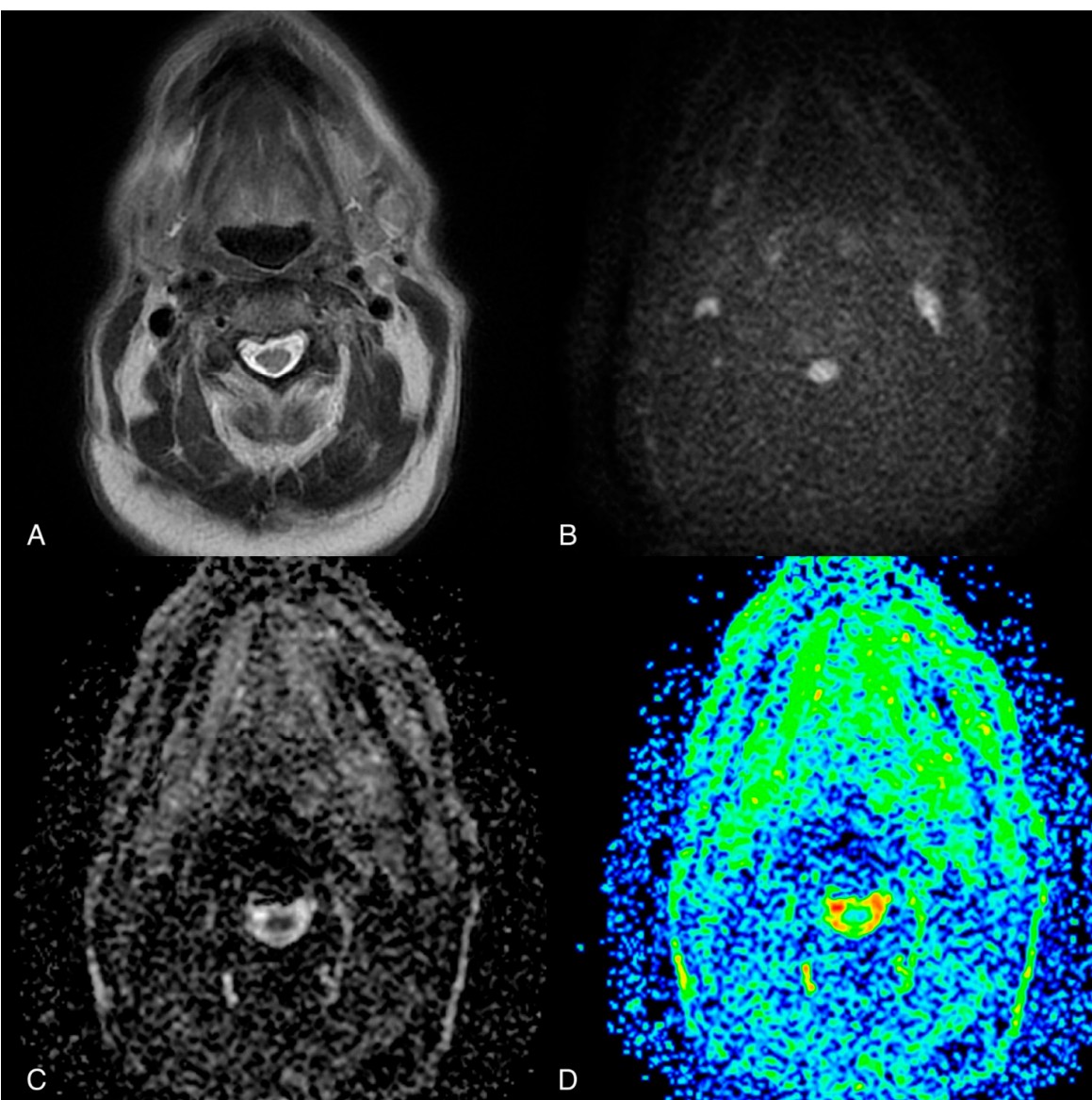

**Figure 2.** Evaluation of the malignant node (station VC) in a 78-year-old male patient with oropharyngeal carcinoma using MR images. The images include T2-weighted sequence (**A**), DWI sequence (**B**), and ADC map in grayscale (**C**) and colorimetric scale (**D**).

*2.4. Statistical Analysis*

The statistical analysis presented in the study aimed to identify the significant parameters that could predict the presence of pathological lymph nodes in patients with cancer. The analysis consisted of two parts: internal validation and external validation. In the internal validation, we used a training population to identify the parameters that were significantly associated with node malignancy. We conducted a univariate analysis using the Chi-Square test to examine the relationship between various characteristics and node malignancy. Next, we performed a multivariate analysis using logistic regression to determine the significant parameters amongst the parameters that resulted significant at univariate analysis. Finally, we generated an ROC curve, which is a graphical representation of the sensitivity and specificity of the prediction model.

Next, we conducted an external validation of the model, applying the significant parameters identified in the internal validation to an external independent dataset. Additionally, in this case, an ROC curve was generated.

## 3. Results

Table 1 presents the summary of the statistics for a sample of patients and their lymph nodes, wherein a total of 43 patients and 49 lymph nodes were analyzed.

**Table 1.** The table contein information about our study's patients. This includes the total number of patients enrolled in the study, the number of patients excluded, the age range of the partecipants and the tumor biology.

| | |
|---|---|
| **Total number of enrolled patients** | 55 |
| **Number of excluded patients** | 12 |
| **Age range** | 55–81 years old (21 male and 22 female) |
| **Tumor biology** | 43 squamous cell carcinoma; 7 Non-Hodgkin lymphoma; 3 adenocarcinoma; 1 carcinoma ex pleomorphic adenoma; myoepithelial carcinoma |

*3.1. Training Population*

Univariate analysis (Chi-Square test) identified six parameters that significantly correlated with the presence of pathological nodes: Dmax (maximum diameter), ROI.p.DWI (DWI Region of Interest in the central area of the lymph node), ROI.p.ADC (ADC Region of Interest in the subcapsular area of the lymph node), SF.c.DWI (DWI 3D Region of Interest in the central area of the lymph node), and SF.p.DWI (DWI 3D Region of Interest in the subcapsular area of the lymph node).

Multivariate analysis (logistic regression analysis) was then conducted to reduce the significant parameters, which revealed that only Dmax and ROI.p.DWI were statistically significant ($p = 0.030$ and $p = 0.016$, respectively). The logistic regression model had an R2 value of 0.791, indicating that the model explained 79.1% of the variability in the dependent variable (presence of pathological nodes).

The ROC curve generated from the logistic regression model had an AUC of 0.944 (95% CI, 0.85–0.98), indicating that the model had good discriminative power. The model had a sensitivity of 92.3%, specificity of 50%, and an accuracy of 82.3% (Figure 3).

*3.2. Validation Population*

In the external validation, the significant parameters in the training population (Dmax and ROI.p.DWI) were evaluated using binary logistic regression analysis. The ROC curve generated from the logistic regression model had an AUC of 0.972 (95% CI: 0.904–1), indicating that the model had excellent discriminative power. The model had a sensitivity of 91.6%, specificity of 83.3%, and an accuracy of 88.8% (Figure 4).

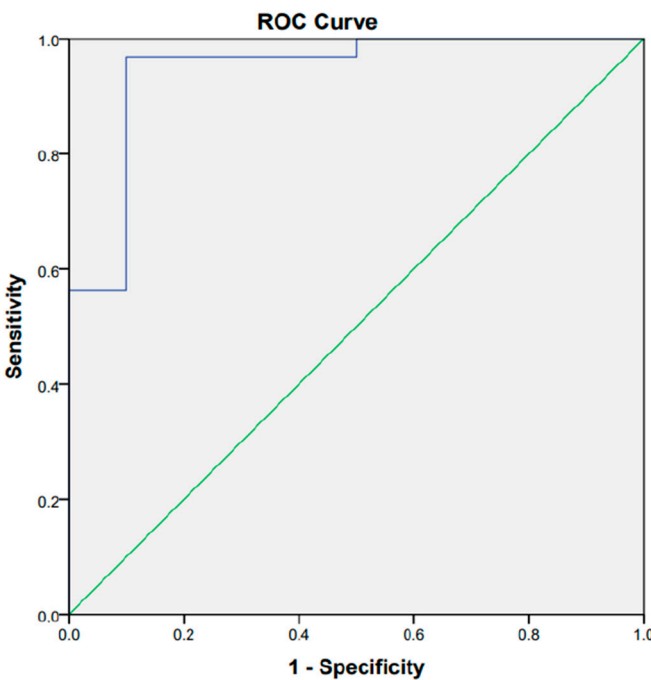

**Figure 3.** ROC curve was generated from the binary logistics regression. The blue line represents our discriminant ( Dmax and ROL.p DWI): AUC resulted to be 0.944. While the green line is the random classifier (AUC 0.5).

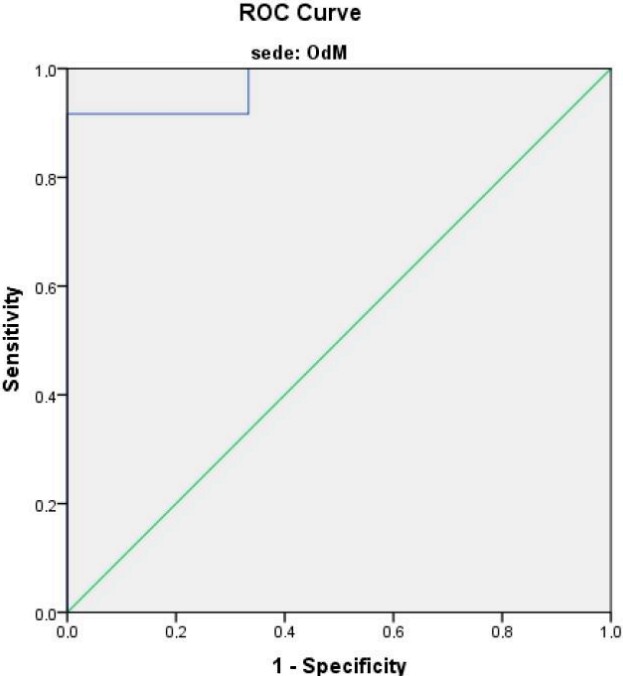

**Figure 4.** Evaluation Fig. 3 ROC curve was generated from the binary logistics regression. The blue line represents our discriminant ( Dmax and ROL.p DWI): AUC resulted to be 0.944. While the green line is the random classifier (AUC 0.5).

## 4. Discussion

It is crucial in diagnosing, determining the stage, developing a treatment plan, and monitoring patients with head and neck cancer, to accurately predict the status of their cervical lymph nodes. However, none of the morphological features, such as size, shape,

or the presence of necrosis, can be fully relied upon, making it challenging to differentiate between malignant and benign lymph nodes [26].

Assessing tissue properties non-invasively is possible with Permeability Magnetic Resonance Imaging (DW-MRI), making it a useful imaging modality. DWI has been explored as a potential cancer biomarker to differentiate between benign and malignant tumors, and this study aimed to determine if DWI can differentiate between benign and malignant lymph nodes by comparing their ADC values [27].

DWI is a non-invasive imaging technique that assesses tissue properties by comparing the movement of water molecules in various tissues [28–31]. This method provides both quantitative and qualitative data, making it useful for determining tissue type. Initially used in neuroimaging [32], DWI has now been applied to the diagnosis and treatment of various clinical diseases [33–36]. While DWI has been widely studied for various applications, it is mainly used in the evaluation of lymph nodes (LNs) in patients with head and neck squamous cell carcinoma (HNSCC). DWI has been found to be useful in diagnosing HNSCC, planning treatment, evaluating response, and improving patient care. Studies have also shown that there are significant differences in DWI detection and ADC map patterns between normal and metastatic LNs in HNSCC patients [37–41].

The conventional method for detecting metastasis involves looking for lymph nodes with a diameter of more than 10 mm in short-axis on CT and MRI images. However, determining an appropriate size for lymph nodes is challenging as their size can vary based on their location in the neck, and they may not always enlarge in response to small metastatic deposits [42].

The detection of small lymph nodes with DWI has only been explored in a few studies. Our study was conducted prospectively and cross-referenced the ADC map of the lymph node with its corresponding histological sample to ensure accuracy and consistency in equipment and examination [43–46]. Regions of interest (ROIs) were manually created over the entire lymph node in each section and mean ADC values were determined. Other studies, such as the work by Vandecaveye et al. [39,47], used whole node-wide ROIs and showed that ADC values in metastatic sub-centimeter lymph nodes were lower than those in benign lymph nodes. Similarly, Dirix et al. [48] used freehand-drawn ROIs across the entire node and found that ADC values in sub-centimeter metastatic lymph nodes were lower than in benign lymph nodes. Most of the previously studied research relied on using ROIs for the entire node.

In a limited number of studies, the size of the region of interest (ROI) has been shown to impact ADC values. Bilgil et al. [49–51] found that the size of the ROI affected ADC values in brain lesions, and the same was discovered for breast lesions in a study by Zhang et al. [52].

The impact of ROI size on ADC values has been studied in a limited number of research studies. In some studies, smaller ROIs were found to result in inconsistent ADC values, particularly if the ROI volume was smaller than the voxel size. However, other studies found that the size of the ROI did not impact ADC values. Herneth et al. [53] found that smaller ROIs can cause erratic ADC values. Conversely, Sun et al. [54] found that ADC measures were unaffected by the size of the ROI in their study comparing pleomorphic adenoma, Warthin tumor, and normal parotid parenchyma. Additionally, high ADC values in benign lymph nodes have been linked to a fatty hilum, while malignant infiltration of the hilum leads to lower ADC values in the central section of metastatic lymph nodes [55]. To test this hypothesis, we positioned a small ROI inside the node's core and drew a larger ROI around the entire node. Necrotic portions of the lymph nodes have higher ADC values, but these areas are too small to be seen on T1 and T2 sequences.

In diagnosing head and neck squamous cell carcinoma (HNSCC), the risk of necrosis and micro necrosis in metastatic lymph nodes increases as the nodes grow in size. Smaller lymph nodes, those measuring less than a centimeter, would have a lower risk of necrosis. Our study found that the ADC (apparent diffusion coefficient) value measured in the center region of the node was lower than that measured in the entire node's ROI. We also found

that using the ROI of the center region of the node produced a higher AUC (area under the curve), sensitivity, and specificity for detecting metastasis compared to using the ROI of the complete lymph node [47].

The use of multiple DWI features to evaluate the most crucial values for lymph node staging has not been thoroughly explored in the studies reviewed. One study found that ADC values alone provide no information about the grade of LN metastases [56]. Additionally, no studies support the idea of using DWI as the unique technique for evaluating head and neck lymph node status. However, one study found no statistically significant difference in the ADC values of metastatic lymph nodes between patients with good and poor prognoses for HNSCC, contrasting the findings of other studies [57]. DWI images of head and neck cancer nodes may have motion artifacts and eddy current distortions, which can affect the accuracy of measurements, such as ADC values. Correcting these errors can improve image quality and ensure accurate measurements.

However, there [58] may also be spatial errors in DWI images, such as gradient nonlinearity, spatial distortion, and varying image scaling factors, that can impact diffusion tensor imaging (DTI) metrics, such as fractional anisotropy and mean diffusivity. The intensity of the diffusion weighted MRI signal is described by the Stejskal–Tanner equation. It assumes that the gradients are uniform throughout the sample. However, this assumption is rarely met in clinical or research MRI scanners. As a result, the equation is only applicable to a specific scenario where the gradients are homogeneous. The presence of non-uniformities in the magnetic field gradients can result in significant artifacts in diffusion imaging. In addition to causing image warping, nonlinearities in the imaging gradients can also generate errors in the direction and magnitude of the diffusion encoding that vary spatially. Diffusion tensor imaging (DTI) is an MRI modality that allows for the investigation of tissue microstructure. However, its reliability is dependent on the performance of diffusion-sensitizing gradients, which can suffer from spatial nonuniformity in most clinical MRI scanners. To test the impact of this nonuniformity, two diffusion phantoms were simulated with varying levels of noise. Two methods were used to calculate the tensors: assuming a spatially constant b-matrix (standard DTI) and applying the b-matrix spatial distribution in the DTI (BSD-DTI) technique. The results showed that diffusion gradient inhomogeneity leads to overestimation of the largest eigenvalue, underestimation of the smallest one, and therefore overestimation of fractional anisotropy, similar to the effect of noise but which cannot be corrected by increasing SNR [59–62].

These corrections are crucial to ensure accurate and reliable DWI images and DTI metrics in head and neck cancer nodal imaging. It is important to note that this study has limitations and further research is needed to explore this topic in more detail [63,64].

The patients' prognosis can be affected by the degree of differentiation of HNSSC. In order to properly treat HNSCC, doctors need to know the tumor's histological grade. Poorly differentiated or moderately differentiated HNSCC is also a grading category. Regional and distant metastases are more common in patients with poorly differentiated HNSCC [65].

According to the studies of Meada et al. [48], there is no noticeable difference in ADC values between the different histologic grades of HNSCC in the primary tumor location. On the other hand, Yun et al. [66,67] found a difference in ADC values between moderately and poorly differentiated HNSCC at the primary site, however, this difference was only significant at a certain b-value (b = 2000). Sumi et al. [55] suggested that the ADC values in the metastatic lymphatic vessels of HNSCC patients with good prognosis and locally advanced disease are higher than those with poorly differentiated HNSCC.

## 5. Conclusions

In conclusion, the study found that MR displacement imaging and ADC values are a non-invasive way to assess metastatic lymph nodes in patients with head and neck cancer. DWI provides quantifiable data that can be used to establish ADC value criteria to determine if nodal lesions are benign or malignant.

A larger ROI has a greater impact on the nodes' ADC. The sensitivity, specificity, and AUC of a narrower ROI for recognizing metastases are all much greater compared to the ADC value of the whole node. ADC levels in metastatic lymph nodes are negatively associated with the histopathological grade of the primary tumor. In conclusion, diffusion imaging may serve as a valuable tool in the evaluation of cervical lymphadenopathies.

**Author Contributions:** Conceptualization, A.R. and C.L.; Methodology, A.R.; Software, G.C. and V.N.; Validation, V.C., C.L. and S.C.; Formal analysis, V.N.; Investigation, G.M.R. and S.C.; Resources, L.G., P.M.P. and G.M.R.; Data curation, V.N.; Writing—original draft, M.P.B. and L.G.; Writing—review & editing, G.C.; Visualization, M.P.B., M.F., R.Z. and S.C.; Supervision, M.P.B., A.R., C.L. and S.C.; Project administration, G.C., V.N. and S.C.; Funding acquisition, L.G. All authors have read and agreed to the published version of the manuscript.

**Funding:** This research received no external funding.

**Informed Consent Statement:** Informed consent was obtained from all subject involved in the study, according to the guidelines of the SIRM.

**Data Availability Statement:** The data are available upon request.

**Conflicts of Interest:** The authors declare no conflict of interest.

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
