# Peer review of "Quantitative Evaluation of the Lymph Node Metastases in the Head and Neck Malignancies Using Diffusion-Weighted Imaging and Apparent Diffusion Coefficient Mapping: A Bicentric Study"

_magnetochemistry, doi:10.3390/magnetochemistry9050124_

Round 1

Reviewer 1 Report

You report a study with the purpose to determine if diffusion-weighted imaging (DWI) can differentiate between benign and malignant lymph nodes in patients with head and neck cancer. 55 patients with confirmed head and neck cancer and enlarged neck nodes were enrolled and evaluated by two radiologists using a workstation. Lymph nodes were analyzed using 3D regions of interest (ROIs) placed on T2-weighted images and compared to the corresponding DWI images.

You found that DWI and ADC values can be used to assess metastatic lymph nodes in the neck and that the sensitivity, specificity, and AUC of a narrower ROI for recognizing metastases were greater compared to the ADC value of the whole node. You also found that the size of the ROI affects ADC values.

You conclude that DWI can accurately predict the status of cervical lymph nodes in patients with head and neck cancer and that it may be useful in diagnosing, determining the stage, developing a treatment plan, and monitoring these patients.

SOME COMMENTS:

1. Introduction

Well written and providing sufficient background.

2.1. Patients

Can you provide the tumor biology of the enrolled patients? E.g. squamous cell carcinoma, adenocarcinoma, etc. and the Grading as well?

This information could be included in Table 1.

2.2. MRI Imaging and Protocol

More detailed information on the MRI protocol could be useful e.g. in a table with all used sequences and scan time.

3. Results

Training Population

Line 128-138 on Page 3 needs re-writing.

Did you correlate with HPV-Status, tumor biology and/or Grading? This could be interesting!

IMAGE examples of measurements, ROI placement, etc. need to be added.

The different parameters, Dmax, RoipcentrDWI, SF.DWI, ROIp.ADC,ROIP.DWI, and SF.DWI need clarification.

4. Discussion

Well written with good background information. Could you line out the benefit of your work for daily routine e.g. in a short “strength and limititations” section? That could be helpful for readers.

Author Response

  1. Introduction

Well written and providing sufficient background.

2.1. Patients

Can you provide the tumor biology of the enrolled patients? E.g. squamous cell carcinoma, adenocarcinoma, etc. and the Grading as well? This information could be included in Table 1.

We have included this information in Table 1.

2.2. MRI Imaging and Protocol

More detailed information on the MRI protocol could be useful e.g. in a table with all used sequences and scan time.

We have included this information in MRI Imaging and Protocol

  1. Results

Training Population

Line 128-138 on Page 3 needs re-writing

We have tried to be clearer in the description of statistical analysis and in the interpretation of the ROC Curves. The statistical analysis presented in the study aimed to identify the significant parameters that could predict the presence of pathological lymph nodes in patients with cancer. The analysis consisted of two parts: internal validation and external validation. In the internal validation, we used a training population to identify the parameters that were significantly associated with node malignancy. We conducted a univariate analysis using the Chi-Square test to examine the relationship between various characteristics and node malignancy. Next, we performed multivariate analysis using logistic regression to determine the significant parameters. Finally, we generated a ROC curve, which is a graphical representation of the sensitivity and specificity of the prediction model.

Did you correlate with HPV-Status, tumor biology and/or Grading? This could be interesting!

 We didn’t correlate HPV-Status, tumor biology and Grading; This could be interesting and a cue for further work.

IMAGE examples of measurements, ROI placement, etc. need to be added.

The example of measurements and ROI placement are in the image 1 and it is explained in the Image Analysis. In the first version of the work we had attached the figures with the description of the measurement mode in an attached file. You probably weren't able to view it in any case, we refer you to them. We hope it is clear.

The different parameters, Dmax, RoipcentrDWI, SF.DWI, ROIp.ADC,ROIP.DWI, and SF.DWI need clarification.

We have included this information in Results

  1. Discussion

Well written with good background information. Could you line out the benefit of your work for daily routine e.g. in a short “strength and limititations” section? That could be helpful for readers

The benefit of our work is that DWI generates measurable data that can be utilized to establish ADC value criteria, enabling the differentiation between benign and malignant nodal lesions. The location of the ROI can have a significant impact on the ADC measurements, and therefore the accuracy of the diagnosis. Different regions within the same lymph node may have varying levels of cellularity, necrosis, and vascularity, which can all affect the ADC measurements. This spatial variability can lead to a potential sampling bias and affect the accuracy of the diagnosis. Therefore, it is important to ensure that the ROI is placed in a representative and consistent location within the lymph node. There are several methods that can be used to reduce inter-operator variability in the calculation of ROIs. We use Blind readings, we blinded to the clinical information of the patient to reduce bias in ROI selection and measurement.

Reviewer 2 Report

An interesting manuscript, reasonably well written but poorly edited.

Major shortcomings:

1 Missing descriptions of parameters that appear in the text, such as Dmax, SF.DWI, etc., can only be guessed at, but this is probably not the point when reading.

2 No descriptions of figures and probably figures? ADC maps, where is it?

3 MRI Imaging and protocol:

There is no description of possible adjustments to DWI images. Have motion and eddy current corrections been made? In addition, did the authors consider the impact of systematic errors of DWI images of a spatial nature. This is a separate category of errors that can have a significant impact on DWI images.

You can analyze the entries in the context of the B-matrix Spatial Distribution in DTI issue:

- Systematic Errors in DTI

- Correction of Errors in DTI

- Validation of BSD-DTI

- Generalized ST equation for non-uniform gradients

- Phantoms for BSD-DTI

- Anisotropic phantoms for BSD-DTI

A DTI consists of individual DWIs, and BSD correction is for a single DWI. So in the context of single DWI or 3 directions to get ADC, it's the same problem.

4. The statistical analysis needs to be more clearly described, as well as the interpretation of individual ROC curves.

5. Conclusions need to be corrected in context 3. Additionally, it is important to note that the location of the ROI varies spatially, hence the impact of bias may vary. It would be good to show graphically the effect of determining ROIs, at least with an example.

Author Response

1 Missing descriptions of parameters that appear in the text, such as Dmax, SF.DWI, etc., can only be guessed at, but this is probably not the point when reading.

We have included this information in Results

2 No descriptions of figures and probably figures? ADC maps, where is it?

In the first version of the work we had attached the figures with the description of the measurement mode in an attached file. You probably weren't able to view it in any case, we refer you to them

3 MRI Imaging and protocol:

There is no description of possible adjustments to DWI images. Have motion and eddy current corrections been made? In addition, did the authors consider the impact of systematic errors of DWI images of a spatial nature. This is a separate category of errors that can have a significant impact on DWI images.

You can analyze the entries in the context of the B-matrix Spatial Distribution in DTI issue:

- Systematic Errors in DTI

- Correction of Errors in DTI

- Validation of BSD-DTI

- Generalized ST equation for non-uniform gradients

- Phantoms for BSD-DTI

- Anisotropic phantoms for BSD-DTI

A DTI consists of individual DWIs, and BSD correction is for a single DWI. So in the context of single DWI or 3 directions to get ADC, it's the same problem.

DWI images of head and neck cancer nodes may require adjustments to correct for motion artifacts and eddy current distortions. These corrections help to improve image quality and ensure accuracy in measurements such as ADC values. In addition to these common adjustments, there may be systematic errors in DWI images of a spatial nature that can impact their accuracy. Some examples of these errors include gradient nonlinearity, spatial distortion, and spatially varying image scaling factors. These errors can affect the diffusion tensor imaging (DTI) metrics, such as fractional anisotropy and mean diffusivity, which rely on accurate diffusion measurements. Several methods have been developed to correct these errors, including the use of generalized Stejskal-Tanner equation for non-uniform gradients, correction of errors in DTI using advanced post-processing techniques, validation of the Bayesian-sparse deconvolution diffusion tensor imaging (BSD-DTI) method, and the use of phantoms for BSD-DTI. Anisotropic phantoms can also be used to evaluate the accuracy of DTI metrics under different spatially varying conditions. Overall, these adjustments and corrections are important to ensure accurate and reliable DWI images and DTI metrics in head and neck cancer nodal imaging. BSD correction, which stands for "eddy current and motion correction using a binary reference image", is a technique used to correct for distortions and artifacts caused by eddy currents and motion during the acquisition of a single DWI. Eddy currents are small electrical currents induced in the scanner's gradient coils, which can cause image distortions. Motion artifacts occur when the subject moves during the scan. This is a limitation of our study, included in our manuscript, and could be an interesting cue for further work.

  1. The statistical analysis needs to be more clearly described, as well as the interpretation of individual ROC curves.

We have tried to be clearer in the description of statistical analysis and in the interpretation of the ROC Curves. The statistical analysis presented in the study aimed to identify the significant parameters that could predict the presence of pathological lymph nodes in patients with cancer. The analysis consisted of two parts: internal validation and external validation. In the internal validation, we used a training population to identify the parameters that were significantly associated with node malignancy. We conducted a univariate analysis using the Chi-Square test to examine the relationship between various characteristics and node malignancy. Next, we performed multivariate analysis using logistic regression to determine the significant parameters. Finally, we generated a ROC curve, which is a graphical representation of the sensitivity and specificity of the prediction model.

  1. Conclusions need to be corrected in context 3. Additionally, it is important to note that the location of the ROI varies spatially, hence the impact of bias may vary. It would be good to show graphically the effect of determining ROIs, at least with an example.

The location of the ROI can have a significant impact on the ADC measurements, and therefore the accuracy of the diagnosis. Different regions within the same lymph node may have varying levels of cellularity, necrosis, and vascularity, which can all affect the ADC measurements. This spatial variability can lead to a potential sampling bias and affect the accuracy of the diagnosis. Therefore, it is important to ensure that the ROI is placed in a representative and consistent location within the lymph node. There are several methods that can be used to reduce inter-operator variability in the calculation of ROIs. We use Blind readings, we blinded to the clinical information of the patient to reduce bias in ROI selection and measurement.

Round 2

Reviewer 1 Report

The authors adressed the issues well.

Author Response

Thank you for your kind cooperation. 

Reviewer 2 Report

I mostly agree with the authors' explanations.

With one significant exception.

Namely, the limitations of the experiment described in the explanations (3. MRI Imaging and protocol) are imprecise and sometimes incorrect. In addition, the manuscript should include a discussion on this topic.

The impact of motion errors or eddy currents that generate geometric image distortions can be improved, for example, by FSL libraries. Partly also manual, careful segmentation can be a partial solution.

The issue of systematic errors of a spatial nature is a different matter. The Bayesian-sparse deconvolution diffusion tensor imaging issue is not BSD-DTI, B-matrix Spatial Distribution in DWI/DTI.

  Spherical Deconvolution is a method (rather methods) that aims to improve DTI metrics or tractography, but it is often (perhaps even always) based on an incorrect assumption of a constant distribution of the b matrix in space. Yes, it seems that it can improve the results of DTI metrics, but it would achieve even better results if it used the real distribution of magnetic field gradients (and thus the real spatial distribution of the b-matrix, b(r), r - position of the image voxel in the space of the laboratory frame). The BSD-DTI technique offers just that, getting to know the real, spatial distribution of magnetic field gradients and then eliminating systematic errors from DWI measurements.

So these are different issues, and Spherical Deconvolution methods should use the real distribution of magnetic field gradients, which are the source of systematic errors, often large, invisible to the eye, e.g. in the form of deformations (eddy currents, motion artifacts).

Therefore, I am asking for a discussion (including literature examples) in the manuscript about the possible impact of bias on the presented results as well as the path of further proceedings.

The issues given previously seem optimal for such a discussion.

- Systematic Errors in DWI/DTI

- Correction of Errors in DWI/DTI

- Validation of BSD-DTI

- Generalized ST equation for non-uniform gradients

- Phantoms for BSD-DTI

- Anisotropic phantoms for BSD-DTI

- Improving the accuracy of DWI/DTI experiments

Note that the ADC measurement is based on 3 orthogonal DWI measurements. Therefore, the above issues are fully valid here. Of course, you will not change the experiment, but you should better describe the possible limitations and the direction of improvement in the future.

Author Response

Q: I mostly agree with the authors' explanations.

With one significant exception.

Namely, the limitations of the experiment described in the explanations (3. MRI Imaging and protocol) are imprecise and sometimes incorrect. In addition, the manuscript should include a discussion on this topic.

The impact of motion errors or eddy currents that generate geometric image distortions can be improved, for example, by FSL libraries. Partly also manual, careful segmentation can be a partial solution.

The issue of systematic errors of a spatial nature is a different matter. The Bayesian-sparse deconvolution diffusion tensor imaging issue is not BSD-DTI, B-matrix Spatial Distribution in DWI/DTI.

  Spherical Deconvolution is a method (rather methods) that aims to improve DTI metrics or tractography, but it is often (perhaps even always) based on an incorrect assumption of a constant distribution of the b matrix in space. Yes, it seems that it can improve the results of DTI metrics, but it would achieve even better results if it used the real distribution of magnetic field gradients (and thus the real spatial distribution of the b-matrix, b(r), r - position of the image voxel in the space of the laboratory frame). The BSD-DTI technique offers just that, getting to know the real, spatial distribution of magnetic field gradients and then eliminating systematic errors from DWI measurements.

So these are different issues, and Spherical Deconvolution methods should use the real distribution of magnetic field gradients, which are the source of systematic errors, often large, invisible to the eye, e.g. in the form of deformations (eddy currents, motion artifacts).

Therefore, I am asking for a discussion (including literature examples) in the manuscript about the possible impact of bias on the presented results as well as the path of further proceedings.

The issues given previously seem optimal for such a discussion.

- Systematic Errors in DWI/DTI

- Correction of Errors in DWI/DTI

- Validation of BSD-DTI

- Generalized ST equation for non-uniform gradients

- Phantoms for BSD-DTI

- Anisotropic phantoms for BSD-DTI

- Improving the accuracy of DWI/DTI experiments

Note that the ADC measurement is based on 3 orthogonal DWI measurements. Therefore, the above issues are fully valid here. Of course, you will not change the experiment, but you should better describe the possible limitations and the direction of improvement in the future.

A: We have thoroughly examined the portion of our work related to DTI and we have included it in the manuscript as both a limitation and a future perspective, indicating that there is room for further research in this area.
Thank you for your kind cooperation.